# The Copula Application for Analysis of the Flood Threat at the River Confluences in the Danube River Basin in Slovakia

Veronika Bačová Mitková *, Dana Halmová, Pavla Pekárová and Pavol Miklánek

Slovak Academy of Sciences, Institute of Hydrology, Dúbravská Cesta 9, 841 04 Bratislava, Slovakia
* Correspondence: mitkova@uh.savba.sk; Tel.: +421-232293503

**Abstract:** In hydrological practice, individual elements of the hydrological cycle are most often estimated and evaluated separately. Uncertainty in the size estimation of extrema discharges and their return period can affect the statistical assessment of the significance of floods. One example is the simultaneous occurrence and joining of extremes at the confluence of rivers. The paper dealt with the statistical evaluation of the occurrence of two independent variables and their joint probabilities of occurrence. Bivariate joint analysis is a statistical approach for the assessment of flood threats at the confluence of rivers. In our study, the annual maximum discharges monitored on four selected Slovak rivers and their tributaries represent the analyzed variables. The Archimedean class of copula functions was used as a set of mathematical tools for the determination and evaluation of the joint probability of annual maximal discharges at river confluences. The results of such analysis can contribute to a more reliable assessment of flood threats, especially in cases where extreme discharges occur simultaneously, increasing the risk of devastating effects. Finally, the designed discharges of the different return periods calculated by using the univariate approach and the bivariate approach for the gauging station below the confluence of the rivers was evaluated and compared.

**Keywords:** flood threat; hydrological extremes; river confluence; bivariate analysis; joint return period; copula function



## 1. Introduction

To assess the significance of extreme events, various statistical methods are used. The theory of probability is one of the most widely used mathematical tools in hydrological practice for the evaluation of event extremity. In hydrological practice, individual elements that characterize the hydrological cycle are most often estimated and evaluated separately, which means that univariate statistical analysis is used. This approach gives satisfactory results when applied to simple systems, for example, where the mainstream does not capture major tributaries [1]. A different situation can arise if rivers with similarly significant overflows form a confluence and extreme events meet at the confluence simultaneously, which can increase the significance of the event or cause a catastrophic situation in the area below the confluence. Artificial interventions in river basins affect or disturb their natural water circulation. Regulations or anthropogenic interventions in river basins often cause changes in their runoff conditions or cause changes in the transformation of flow waves in the riverbed. Such interventions can result in the joint occurrence of flood waves at river confluences. In addition, climate changes also affect the behavior of hydrological extremes along rivers and the interdependent structures between hydrological characteristics. In order to obtain this information, multivariate statistical analysis with a combined cumulative distribution function and a probability density function can be used. A Gaussian probability distribution was used as the first joint distribution function, but the marginal probability distributions had to be normally distributed [2,3]. The limitation of this approach in hydrology is that all variables have the same probability distribution, while the analyzed elements have different marginal distributions. Copula functions are

mathematical tools that can be used to solve the above-mentioned limitation regarding marginal functions in the two-dimensional analysis of hydrological elements. Copula functions express the structure of dependence between random variables, regardless of their marginal distribution. In the context of mentioned factors (climate change, anthropogenic activities), a multivariate statistical approach seems to be a suitable mathematical tool for analyzing changes in the mutual dependence of the natural variables or joint occurrence of extremes. The joint probability of two random variables in contrast to the conditional probability reflects the probability with which the two random variables occur simultaneously [4]. The monograph on the Danube by [5] dealt with the coincidence between flood waves in the Danube River and its selected tributaries. The bivariate analysis confirmed that flood wave genesis in the Danube River basin is very complex. The coincidence of waves caused a flood with a return period of 100 years in 2000 on the Tisa and the Bodrog Rivers. The coincidence of the individual flood waves in the profiles of the Vltava and the Dyje Rivers (Czech Republic) significantly increased the flow return period in basin areas, causing extreme floods in August 2002 [6]. Espinoza et al. [7] dealt with the bivariate analysis of the great flood that occurred in 2012 in the Amazonas River, which was caused by two large simultaneous flood waves. Li et al. [8] studied the effects of the coinciding flood peaks and the impact of the high precipitation for the Dongting Lake region on flood vulnerability.

The Archimedean class of copula functions is very popular in hydrological applications for studying the relations between the individual elements of a hydrological cycle. This class of copulas is popular in empirical applications due to their flexibility, easy construction, and whole suite of closed-form copulas that cover a wide range of dependency structures including comprehensive and non-comprehensive copulas, radial symmetry and asymmetry, and asymptotic tail dependence and independence. Their applications in flood frequency analyses are very often focused on interdependence analyses between flood characteristics such as peak discharge, volume, and duration. The choice of a specific copula depends on the nature of the dataset. De Michele et al. [9] used the Gumbel–Hougaard copula to model the joint distribution of flood peak and volume to check the adequacy of the dam spillway. Zhang and Singh [10,11] applied the bivariate selected copulas and trivariate Gumbel–Hougaard to obtain conditional return periods of flood peak, volume, and duration. Reddy and Ganguli [12] concluded that the Frank copula better fit the joint and conditional return periods of the mentioned flood characteristics compared to the Ali-Mikhail-Haq, Clayton, Gumbel–Hougaard copulas. The Clayton and Gumel–Hougaard copulas were selected as appropriate tools for bivariate analysis of the flood characteristics at the Bratislava Gauging Station on the Danube River in Slovakia [13]. The Clayton and Student-t copulas were selected as appropriate tools for the bivariate analysis of the flood characteristics at the Litija Gauging Station on the Sava River in Slovenia [14]. The Galambos, Gumbel, and Hüsler–Reiss copulas showed the best performance for synoptic and flash floods, while the Frank copula showed the best performance for snowmelt floods for the bivariate modeling of the relationship between the flood peaks and volumes with a focus on flood generation processes [15]. The bivariate analysis of the peak discharge and volumes with the use of various copula families (11 copula functions) on the Danube River showed that most favored the Frank copula rather than the Clayton and Normal copulas [16]. The study of the bivariate dependences and joint probabilities of various hydrological variables of the Morava River (Slovakia) using Archimedean, extreme value (EV), and Archimax copulas was published in Matúš [17]. The Frank or Gumbel copulas were selected for bivariate drought frequency analysis because these copulas well reproduce the upper tail dependence structure between drought duration and severity [18]. Bezak et al. [19] presented the method of determination of rainfall IDF (intensity–duration–frequency) curves using Frank copula and evaluated the results with empirical rainfall thresholds for selected extreme events that occurred in Slovenia.

Chen et al. [20] applied multidimensional copulas to a flood risk analysis of the corresponding peak discharges at selected river confluences and evaluated the X-Gumbel

copula function as an appropriate tool for assessing the joint conditional distribution function and for return periods of joint discharges. Some Archimedean copulas were applied to assess the combined flood occurrence at the confluence of the rivers Rhine and Sieg (Germany) [21]. In Peng et al. [22], the joint probabilities of the annual maximum discharges were investigated using Archimedean and elliptical copulas and compared with the monthly maximum discharges between the mainstream and its tributaries. Fischer et al. [23] determined long synthetic samples of peak–volume pairs using copulas, which were then applied in a multivariate statistical flood frequency analysis considering flood types and impact of the tributary. Joint frequency analysis at the confluences of the Sava River showed the potential risk of underestimating the design discharges estimated by using the traditional one-dimensional analysis of extremes [24]. The analysis of the joint probabilities of flood occurrences at the Danube and Drava confluence showed that the joint probability of an extreme situation is relatively low (0.79%) [25]. The authors of [25] concluded that such a situation occurred in 1966, and it was one of the biggest floods on record.

Therefore, the objective of this study was to apply a bivariate Archimedean copula to analyze the joint probabilities of flood hazard at the Slovak River confluences. The four mainstreams and their tributaries of the Slovak part of the Danube River basin were selected for analysis. Annual maximum discharges from the upper parts of the selected rivers were chosen, and the Gumbel–Hougard copula function was applied to construct the joint distribution. The results of the analysis allow for a comparison and evaluation of the annual maximum discharges with specific return periods T calculated by the univariate flood frequency approach and by the copula approach under the confluences.

## 2. Materials and Methods

### 2.1. Materials

Around 96% of the Slovak territory belongs to the Black Sea drainage area. The Black Sea drainage area represents a part of the basin of the European waterway—the Danube—on the territory of Slovakia and its tributaries: Morava, Váh, Hron, and Ipeľ. The basins of Slaná, Bodva, Hornád, and Bodrog belong to the drainage area of Tisa. The Baltic Sea drainage area (4%) represents the basins of the Poprad and Dunajec Rivers. We selected the four mainstreams of the Slovak Danube River basin and their tributaries in the sections with the least anthropogenic influence (the upper sections of the river) for analysis in the present study. The longest possible data series of the annual maximum discharges were analyzed. Table 1 lists the selected mainstreams, tributaries, gauging stations, and monitored periods. The scheme of the selected rivers and their tributaries is illustrated in Figure 1.

The courses of the selected and analyzed maximum annual discharge pairs are presented in Figure 2. In bivariate flood analysis at the river confluences, the data from gauge stations located on the mainstream immediately above and below the confluence and the gauge station located on the tributary above the confluence were used. We investigated the maximum annual discharge pairs $Q_{maxup}$–$Q_{maxtr}$, where $Q_{maxup}$ is the maximum annual discharge of the mainstream above the confluence, $Q_{maxtr}$ is the maximum annual discharge of the tributary above the confluence, and $Q_{maxdwn}$ represents the maximum annual discharge of the mainstream below the confluence. The Váh River and its tributary, the Belá River, showed the highest differences in the dates when the annual maximum discharges occur (Figure 3a). In addition, the Nitra River and its tributary, the Bebrava River, showed the lowest difference in the dates when the annual maximum discharges occurred (Figure 3b).

The date when the annual maximum discharges between the Váh River and their tributary the Belá River occurs indicates a partly different flood regime between the upper part of the Váh and its tributary, the Belá. Authors of the monograph [26] assumed that the orientations of the various mountains in its path and the depression position of the catchment as well as the different spatial distributions of precipitation might cause the partially different flood regime of the upper part of the Váh and its tributary.

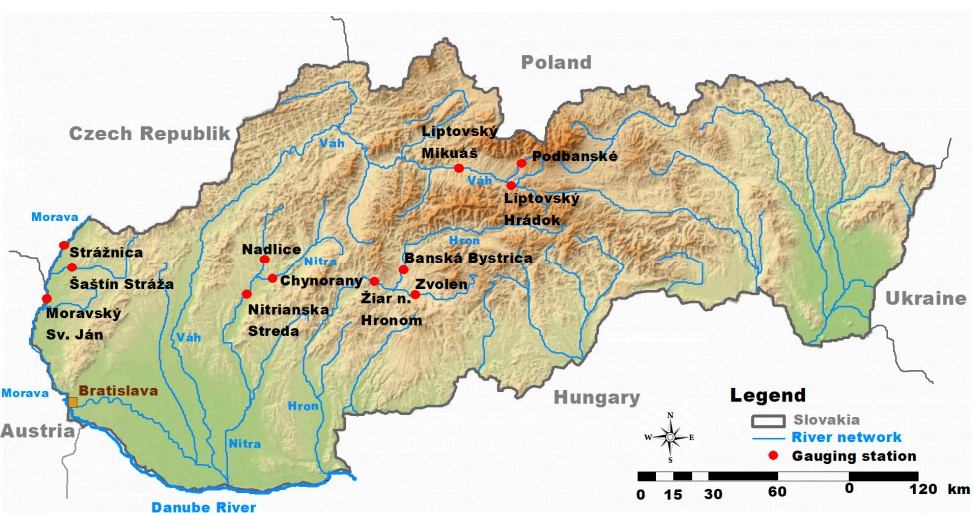

**Figure 1.** Selected river network and gauging stations.

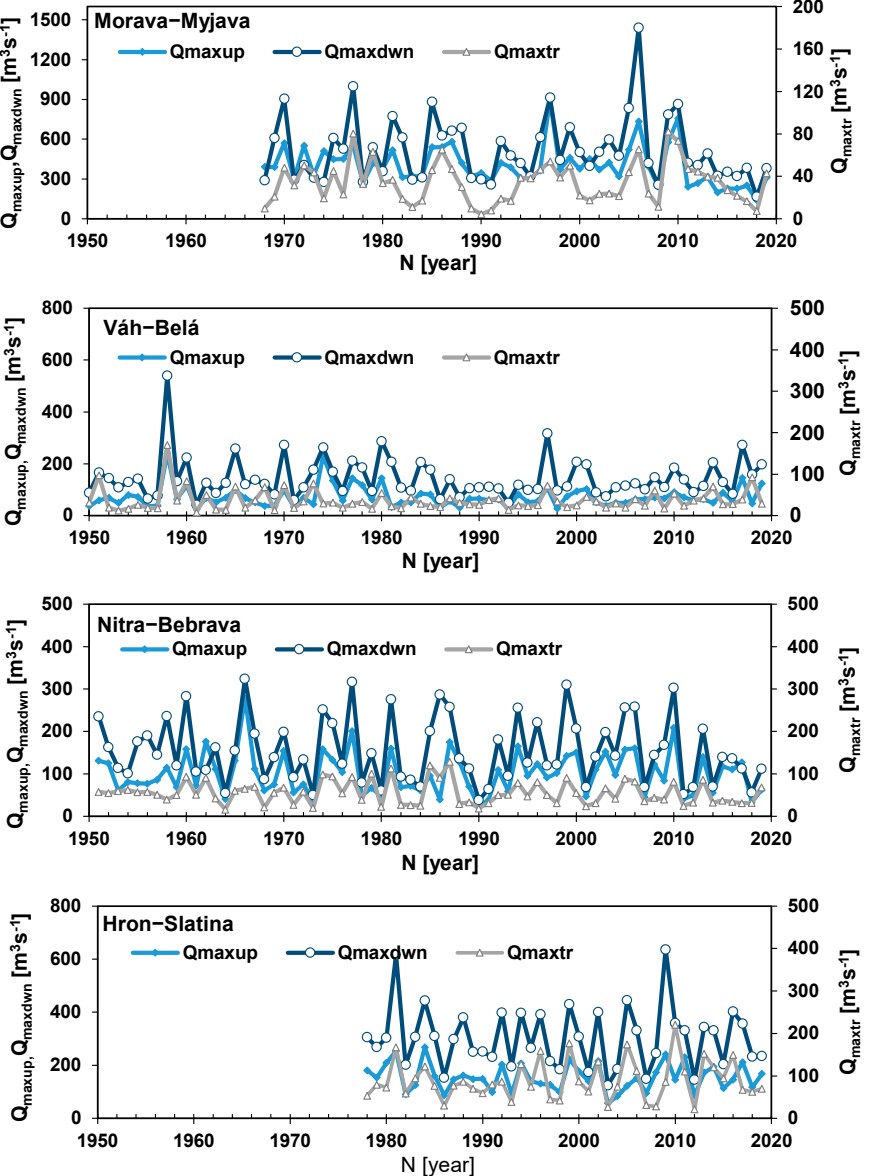

**Figure 2.** Maximum annual discharges for the mainstreams and their tributaries during the analyzed periods: Morava–Myjava, Váh–Belá, Nitra–Bebrava, Hron–Slatina.

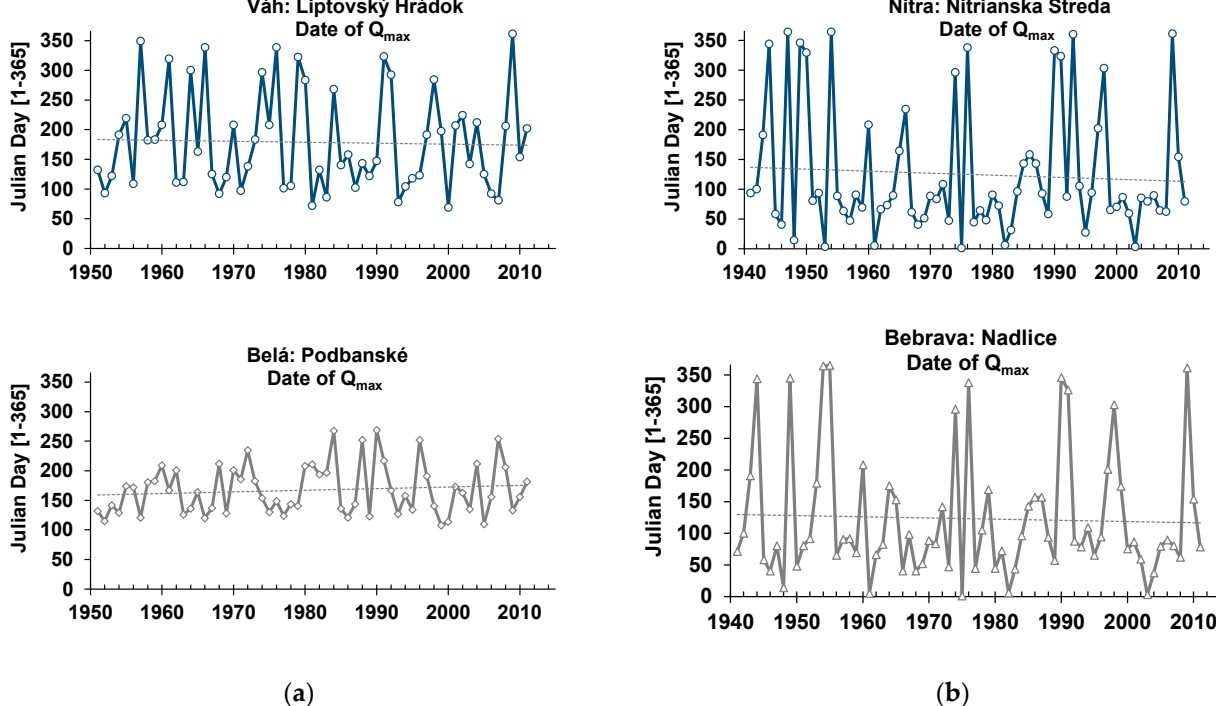

**Figure 3.** Occurrence of the annual maximum discharges in Julian days for (**a**) Váh: Liptovský Hádok (1950–2011) and Belá: Podbanské (1950–2011); (**b**) Nitra: Chynorany (1940–2011) and Bebrava: Nadlice (1940–2011).

**Table 1.** Selected mainstreams and their tributaries, stations, analyzed periods, annual maximum discharge, river kilometer, and basin area.

| River | Gauging Station | Period [Year] | $Q_{max}$ [m³ s⁻¹] | River Kilometer [rkm] | Area [km²] |
|---|---|---|---|---|---|
| Morava | Strážnica (up) | 1968–2019 | 901 | 134.3 | 9146.92 |
| | Moravský Svätý Ján (dwn) | 1968–2019 | 1400 | 67.15 | 24,129.30 |
| Myjava | Šaštín Stráže (tr) | 1968–2019 | 82 | 15.15 | 644.89 |
| Váh | Liptovský Hrádok (up) | 1950–2019 | 240 | 359.3 | 638.38 |
| | Liptovský Mikuláš (dwn) | 1950–2019 | 540 | 343.6 | 1107.21 |
| Belá | Podbanské (tr) | 1950–2019 | 170 | 21.35 | 93.49 |
| Nitra | Chynorany (up) | 1951–2019 | 279 | 106 | 1134.28 |
| | Nitrianska Streda (dwn) | 1951–2019 | 324 | 91.1 | 2093.71 |
| Bebrava | Nadlice (tr) | 1951–2019 | 128 | 6.2 | 598.8 |
| Hron | Banská Bystrica (up) | 1972–2019 | 260 | 175.2 | 1766.48 |
| | Žiar nad Hronom (dwn) | 1972–2019 | 636 | 131.5 | 3310.69 |
| Slatina | Zvolen (tr) | 1972–2019 | 220 | 12.1 | 790.16 |

*2.2. Method*

In our study, we used the Gumbel–Hougaard copula function to perform the statistical bivariate analysis of flood hazards at the selected Slovak River confluences. The Gumbel–Hougaard copula belongs to the Archimedean class of copulas. The relationships between Kendall's coefficient and the generating function show that only the positive dependence structure of the data can be analyzed (e.g., peak–volume or duration–volume) [27–32]. This copula function does not allow for negative dependence and exhibits a strong upper-tail dependence and a relatively weak lower-tail dependence. The Gumbel–Hougard copula may therefore be a suitable choice for dependencies where there is a strong correlation at high values but a weaker correlation at low values. The first step of the bivariate analysis is to identify the marginal distribution. Hydrological variables can have different properties,

so it is necessary to scale the data to variables in the interval [0, 1]. Next, we can separate the marginal behavior and the dependence structure. The joint distribution obtained from the marginal distributions of the uniform variables completely describes the dependence structure of the variables. The mathematical description of the Gumbel–Hougaard copula is listed in Table 2. The copula parameter θ was calculated according a mathematical relationship between Kendall's rank correlation and the generating function φ(t) [33].

**Table 2.** Mathematical description of the applied Gumbel–Hougaard copula.

| Copula Function | C (u, v, θ) | Parameter θ | Kendall's τ | Generator φ(t) |
|---|---|---|---|---|
| Gumbel–Hougaard | $\exp\left[-\left((-\ln u)^\theta + (-\ln v)^\theta\right)^{1/\theta}\right]$ | $[1, \infty)$ | $\frac{\theta-1}{\theta}$ | $(-\ln t)^\theta$ |

The possibilities of statistically testing how well a copula function fits a set of empirical data have been studied in many publications (e.g., [34–38]). Meylan et al. [39] divided the testing into three groups: (1) Based on probability integral transformation; (2) based on the kernel estimation of the copula density; and (3) based on the empirical processes of copulas. There exist several goodness-of-fit tests for comparing the empirical joint probability population and the probability population derived by parametric copulas (e.g., Kolmogorov–Smirnov, Chi-square, Anderson–Darling or Cramér–von Mises). The empirical probability [40–42] is represented in Equation (1):

$$F_{(x,y)} = \frac{\sum_{m=1}^{i} \sum_{l=1}^{i} n_{ml} - 0.44}{N + 0.12} \tag{1}$$

where $N$ is the number of the variables; $j$ and $i$ are ascending ranks of $x_i$ and $y_i$; and $n_{ml}$ is the number of occurrences of the combinations of $x_i$ and $y_j$.

In the hydrological frequency analysis, the return period of the hydrological variable that occurs once in a year can be defined as

$$T = \frac{\mu}{\left(1 - F_{(x)}\right)} \tag{2}$$

where $T$ is the return period; $F(x)$ is the marginal cumulative distribution function; and $\mu$ is an average arrival interval between events. In the frequency analysis of the annual values, $\mu = 1$.

In bivariate statistical analysis, we can use two formulas to calculate the joint return period, depending on whether only one or both of the monitored values exceed or are equal to a certain threshold value that defines the extremity. The joint return period equations of two variables are defined and described in many publications in the following form [43–45]:

$$T_{(x,y)}^{and} = \frac{\mu}{\left(1 - F_{(x)} - F_{(y)} + C\left(F_{(x)}, F_{(y)}\right)\right)} \tag{3}$$

$$T_{(x,y)}^{or} = \frac{\mu}{\left(1 - C\left(F_{(x)}, F_{(y)}\right)\right)} \tag{4}$$

$$T^{or}{}_{(x,y)} \leq \min\left[T_x, T_y\right] \leq \max\left[T_x, T_y\right] \leq T^{and}{}_{(x,y)} \tag{5}$$

where $C(F_{(x)}, F_{(y)})$ is the bivariate joint distribution function expressed as a copula function and $F_{(x)}$ and $F_{(y)}$ are the marginal distribution functions of the variable $X$ and $Y$. The operator of the penetration ($X \geq x$ and $Y \geq y$) is used in Equation (3), and therefore, this formula was used to calculate the joint return period if both investigated quantities exceeded a certain threshold value. Equation (4), on the other hand, works with the operator of the unification ($X \geq x$ or $Y \geq y$). For this reason, it was used to calculate the

joint return period if only one of the monitored values exceeded the given threshold value. These relationships indicate that different combinations of the numbers $x$ and $y$ can cover the same return period.

The conditional return period for $Y$, given $X \geq x$, may be expressed as [44]:

$$T_{(y|X \geq x)} = \frac{\mu}{\left(1 - F_{(x)}\right)\left(1 - F_{(y)} - F_{(x)} + C\left(F_{(x)}, F_{(y)}\right)\right)} \tag{6}$$

The conditional cumulative distribution function of $Y$, given $X \geq x$, can be expressed as

$$F_{(y|X \geq x)} = \frac{F_{(y)} - C\left(F_{(x)}, F_{(y)}\right)}{\left(1 - F_{(x)}\right)} \tag{7}$$

where $C(F_{(x)}, F_{(y)})$ is the copula function of the random variables $X$ and $Y$. An equivalent formula for the conditional return period of $Y \leq y$, given $X \leq x$, can thus be obtained.

The bivariate return period for variables are illustrated using contour lines called isolines. The isolines of the joint "or" return period are the level curves of the G–H copula of interest, while the isolines of the joint "and" return period are the level curves of the survival copula of interest.

## 3. Results

### 3.1. Univariate Statistical Analysis of Flood Hazards

First, the univariate parametric marginal distributions of the annual maximum discharges were identified. In our study, the Johnson SB distribution was evaluated as the most acceptable marginal distribution from the various tested distributions, except for station Nitrianska Streda, where the Weibull probability distribution most acceptable fitted the data. The Johnson SB distribution is a continuous four-parametric distribution defined on a bounded range, and the distribution can be symmetric or asymmetric (Equation (8)). Svanidze and Grigolia [46] recommended this probability distribution as being suitable for annual discharges. This parametric distribution has flexibility in comparison to commonly used distributions such as the Log-normal and the Gamma distributions [47]. The PDF for a variable X that follows an SB PDF can be expressed as [48]:

$$f(x) = \frac{\delta \lambda}{\sqrt{2\pi}(x - \xi)(\xi + \lambda - x)} \, exp(-\frac{1}{2}[\lambda + \delta \ln\left(\frac{x - \xi}{\xi + \lambda - x}\right)]^2) \tag{8}$$

where $\xi < x < \xi + \lambda$, $\lambda$, and $\sigma$ are $>0$, $-\infty < \xi < \infty$, and $-\infty < \gamma < \infty$, respectively.

The parameter $\lambda$ gives the rang;, $\xi$ is the location parameter (lower bound); $\delta$ and $\gamma$ are the shape parameters, and $\gamma = 0$ indicates symmetry.

The Kolmogorov–Smirnov (K–S) test was used to test the assumption that a theoretical distribution follows the empirical discharge magnitudes. A $p$-value at a 5% significance level was used as the criterion for the acceptance of the proposed distribution hypothesis. Empiric distributions, evaluated with the Cunnane [41] formula, were fitted with the selected parametrical cumulative distribution function. Based on the goodness-of-fit test, the calculated RMSE, and a graphical comparison between the monitored data and generated data, the selected parametrical distribution functions were used as the marginal distribution functions for bivariate frequency analysis (Figure 4). The annual maximum discharges estimated using a marginal probability distribution for the selected return period T are listed in Table 3. The selected parametrical distribution function was also used as the marginal distribution function in the joint frequency analysis of flood hazards at selected Slovak River confluences using copulas.

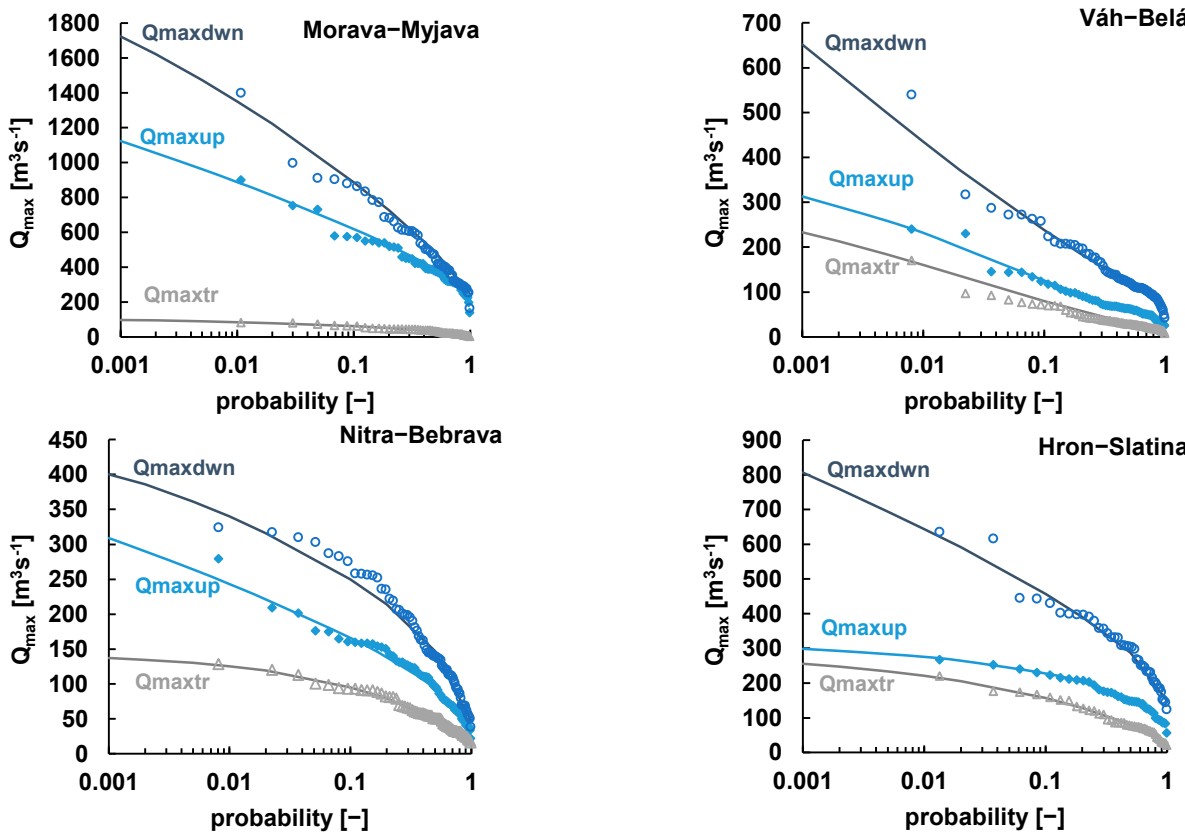

**Figure 4.** Comparison of the empirical exceedance probabilities (points) and the theoretical exceedance probability curves (line) of the maximum annual discharges for the selected mainstreams and tributaries during the analyzed periods.

**Table 3.** Estimated univariate designed discharge $Q_{max}$ for various return periods T and *p*-values of the Kolmogorov–Smirnov test.

| Confluence | Q [m³ s⁻¹] | Distr. | *p* Value | Estimated $Q_T$ [m³ s⁻¹] | | | | | Monitored $Q_{max}$ [m³ s⁻¹] | |
|---|---|---|---|---|---|---|---|---|---|---|
| | | | | Q50 | Q100 | Q200 | Q500 | Q1000 | $Q_{max}$ | T [year] |
| Morava–Myjava | $Q_{maxup}$ | JSB | 0.932 | 815 | 892 | 966 | 1059 | 1127 | 901 | 145 |
| | $Q_{maxtr}$ | JSB | 0.812 | 80 | 85 | 90 | 95 | 98 | 82 | 70 |
| | $Q_{maxdwn}$ | JSB | 0.911 | 1221 | 1351 | 1473 | 1621 | 1723 | 1400 | 160 |
| Váh–Belá | $Q_{maxup}$ | JSB | 0.24 | 200 | 232 | 259 | 290 | 313 | 240 | 160 |
| | $Q_{maxtr}$ | JSB | 0.95 | 136 | 160 | 185 | 213 | 234 | 170 | 160 |
| | $Q_{maxdwn}$ | JSB | 0.87 | 372 | 435 | 499 | 587 | 652 | 540 | 310 |
| Nitra–Bebrava | $Q_{maxup}$ | JSB | 0.92 | 225 | 247 | 268 | 295 | 314 | 279 | 220 |
| | $Q_{maxtr}$ | JSB | 0.91 | 119 | 125 | 130 | 134 | 137 | 128 | 140 |
| | $Q_{maxdwn}$ | Weib. | 0.76 | 322 | 346 | 368 | 386 | 400 | 324 | 60 |
| Hron–Slatina | $Q_{maxup}$ | JSB | 0.97 | 266 | 277 | 285 | 294 | 299 | 268 | 50 |
| | $Q_{maxtr}$ | JSB | 0.83 | 206 | 221 | 234 | 247 | 256 | 220 | 100 |
| | $Q_{maxdwn}$ | JSB | 0.86 | 590 | 643 | 692 | 756 | 806 | 636 | 90 |

### 3.2. Bivariate Statistical Analysis of Flood Hazards at River Confluences Using Gumbel–Hougaard Copula

The Gumbel–Hougaard copula, which is one of the Archimedean copulas, was used as a mathematical tool to calculate the joint probability of the two hydrologic variables at the river confluences. In Feng et al. [49], the authors used three types of time-varying

copula functions to analyze the risk of coinciding, nonstationary floods. The authors of [49] recommended the Gumbel copula rather than the Frank and Clayton copulas.

Correlation analysis showed a strong positive dependence between the monitored annual maximum discharges at the confluences of all rivers. Kendall's rank correlation coefficient ranged between 0.23 and 0.48 (Figure 5a,b). The correlation of the selected combination of hydrological variables showed a statistically significant correlation. The combination of the variables $Q_{maxup}-Q_{maxtr}$ between the Váh River and the Belá River reached the lowest value of the Kendall rank correlation (Figure 5). The created combinations of the hydrological variables were used in the bivariate frequency analysis to investigate how the relationship of the hydrological characteristics may affect the size of extreme hydrological situations.

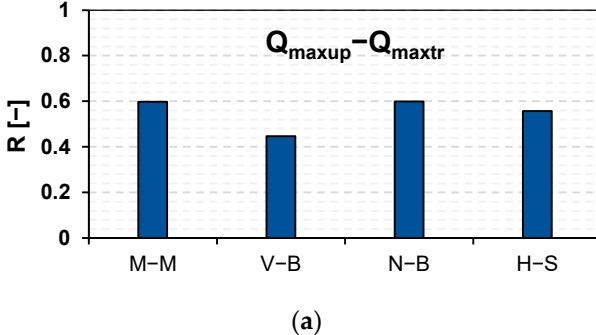
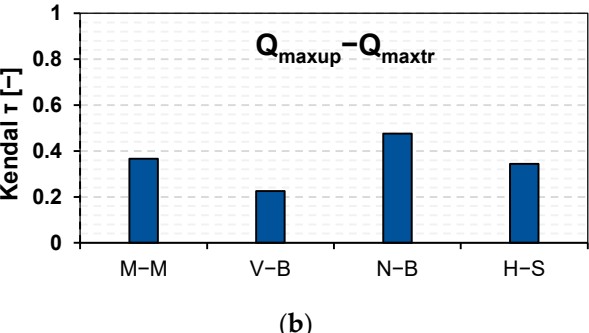

(**a**)　　　　　　　　　　　　　　　　　　　　(**b**)

**Figure 5.** (**a**) Pearson correlation R of the maximum annual discharges and (**b**) Kendall's rank correlation τ of the maximum annual discharges.

The Gumbel–Hougaard copula's parameters are listed in Table 4. The K–S nonparametric goodness-of-fit test was used to assess whether the selected copula function followed the joint empirical distribution. The results show that we cannot reject the hypothesis that the selected copula fits well with the empirical data at a 5% significance level (Table 4). Second, the mean absolute errors (MAE) were used as statistical criteria to determine the level of agreement between the discharges simulated by the copula function and the actual monitored discharges (Table 4). Graphical comparisons of the joint empirical and fitted Gumbel–Hougaard copula of the selected rivers and their tributaries for selected pairs $Q_{maxup}-Q_{maxtr}$ are illustrated in Figure 6a–d. According the above-mentioned criteria, the Gumbel–Hougaard copula was deemed to be a suitable statistical tool to calculate the joint probability distribution of the discharges in our study.

**Table 4.** The Gumbel–Hougaard copula parameters for the selected variable combinations, mean absolute errors (MAE) values, and K–S test.

| Confluence | Pair | Kendall's τ | Parameter Copula | MAE [%] | *p*-Value |
|---|---|---|---|---|---|
| Morava–Myjava | $Q_{maxup}-Q_{maxtr}$ | 0.366 | 1.577 | 4.02 | 0.73 |
| Váh–Belá | $Q_{maxup}-Q_{maxtr}$ | 0.225 | 1.290 | 2.27 | 0.96 |
| Nitra–Bebrava | $Q_{maxup}-Q_{maxtr}$ | 0.476 | 1.908 | 5.94 | 0.052 |
| Hron–Slatina | $Q_{maxup}-Q_{maxtr}$ | 0.366 | 1.567 | 4.94 | 0.78 |

Subsequently, the selected copula function was used to generate 9000 pairs of selected combinations of hydrological variables on the monitored river confluences. The generated pairs were used to determine the joint probability distribution using copulas. Subsequently, the joint return periods of the analyzed pairs of the annual maximum discharges $Q_{maxup}-Q_{maxtr}$ were estimated. Figure 7 illustrates the scatter plot of the monitored annual maximum discharges and values generated by using the Gumbel–Hougard copula. Figure 7 also illustrates the isolines of the joint return periods "or" and "and", which are the level

curves of the Gumbel–Hougaard copula of interest if the variables exceeded the outward bounds and inward bounds, respectively.

The extreme values of the combined discharges' joint return periods were calculated, (i.e., the maximum discharge resulting from combinations of discharges of the mainstreams and their tributaries). These extreme values represent the worst case scenario for flood hazards at these confluences. Table 5 presents a comparison of the estimated T-year designed discharges ($Q_{50}$, $Q_{100}$, $Q_{200}$, $Q_{500}$, and $Q_{1000}$) based on the univariate approach and the copula approach for gauging stations below the selected mainstream river confluences. For the traditional univariate method, the resulting discharge for the selected return period was calculated as a reciprocal of the probability of exceedance.

**Table 5.** Comparison of design discharges ($Q_{50}$, $Q_{100}$, $Q_{500}$, $Q_{1000}$) based on univariate (Uni.) and based on the copula method at the mainstream stations below the confluences $Q_{maxdwn}$.

| Confluence (Station on Mainstream below the Confluence) | Method/Differences | Estimated $Q_T$ [m³ s⁻¹] | | | | |
|---|---|---|---|---|---|---|
| | | $Q_{50}$ | $Q_{100}$ | $Q_{200}$ | $Q_{500}$ | $Q_{1000}$ |
| Morava–Myjava (Morava: Moravský Sv. Ján) | Uni−SB distr. | 1221 | 1351 | 1473 | 1621 | 1723 |
| | copula G–H | 1369 | 1500 | 1623 | 1722 | 1878 |
| | Difference [%] | 12 | 11 | 10 | 6 | 9 |
| Váh–Belá (Váh: Liptovský Mikuláš) | Uni−JSB distr. | 372 | 435 | 499 | 587 | 652 |
| | copula G–H | 446 | 508 | 570 | 651 | 712 |
| | Difference [%] | 20 | 17 | 14 | 11 | 9 |
| Nitra–Bebrava (Nitra: Nitrianska Streda) | Uni−weib. distr. | 322 | 346 | 368 | 386 | 400 |
| | copula G–H | 336 | 354 | 369 | 388 | 401 |
| | Difference [%] | 4 | 2 | 0 | 1 | 0 |
| Hron–Slatina (Hron: Žiar nad Hronom) | Uni−JSB distr. | 590 | 643 | 692 | 756 | 806 |
| | copula G–H | 649 | 699 | 747 | 808 | 853 |
| | Difference [%] | 10 | 9 | 8 | 7 | 6 |

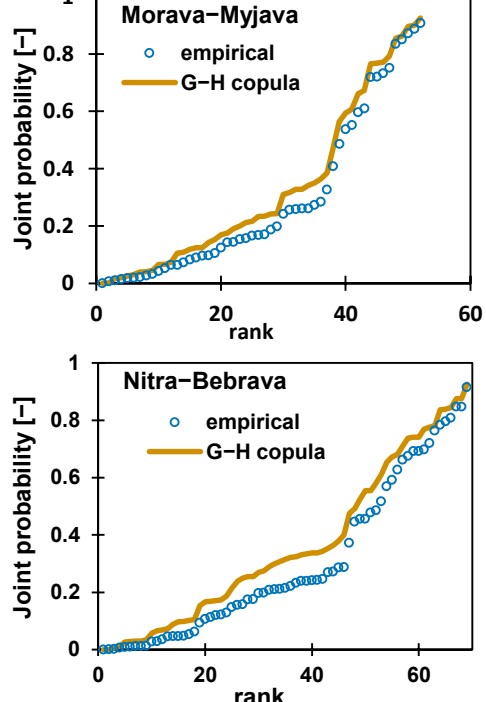
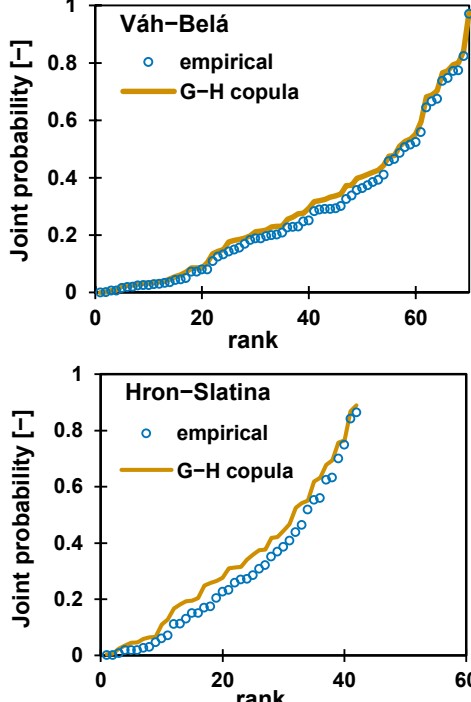

**Figure 6.** The joint empirical probability (points) and Gumbel–Hougaard copula at selected rivers and their tributaries ($Q_{maxup}$−$Q_{maxtr}$).

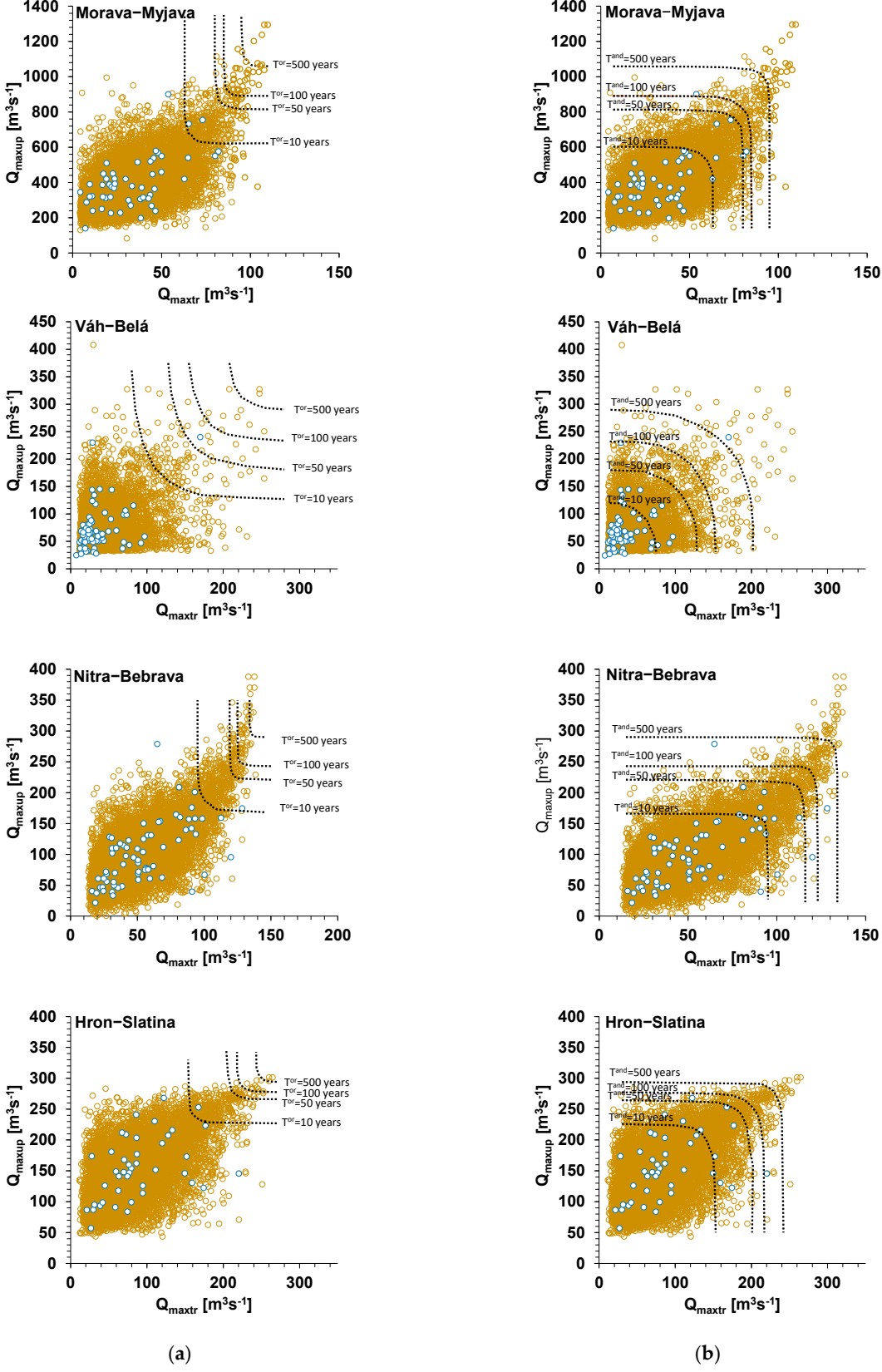

**Figure 7.** Scatter plots of 9000 data pairs generated from Gumbel–Hougaard copula and monitored data of the selected pairs $Q_{maxup} - Q_{maxtr}$. The contours in the return periods (**a**) in the case of "or" ($X \geq x$ or $Y \geq y$, only one investigated variables exceeded a certain threshold value) and (**b**) in the case of "and" ($X \geq x$ and $Y \geq y$, both investigated variables exceeded a certain threshold value).

## 4. Discussion

In the submitted study, the long-term annual maximum discharges at four mainstreams in the Slovak part of the Danube River basin (Morava, Váh, Nitra, and Hron) and their tributaries (Morava, Belá, Nitra, and Slatina) were analyzed. The calculated joint probabilities indicate that coincident flood events are more likely to occur for medium-scale or small-scale flood events. The calculated joint probabilities for large flood events was nearly equal to zero. In following, the extreme values of the joint return periods of the combined discharges were calculated (i.e., the maximum discharge resulting from the combinations of the discharges of the mainstreams and their tributaries). These extreme values represent the worst case scenarios for flood hazards at these confluences. The results of the copula application for analysis of the flood threat at the Slovak River confluences showed that the discharges estimated by a copula in the station below the confluence achieved higher values than the discharges estimated by a univariate approach. The results of the analysis also showed that differences between the discharges estimated by a copula and univariate approach below the confluence decreased with an increase in the return period. For the Váh–Belá River confluence, the differences in annual maximum discharges below the confluence calculated by a copula in comparison to the traditional approach ranged from 20% for 50-year return periods to 11% for a 500-year return period. These relatively high differences between discharges estimated using the copula and univariate approach could indicate different hydrological regimes of the Váh and Belá Rivers for floods with high values of the return period. On the other hand, the relatively low differences between the discharges estimated using the univariate and copula approaches indicate a similar hydrological regime in the mainstream and its tributary.

Catastrophic floods have had and continue to have an important impact on the environment and the economy. Identifying flood risk is very important and difficult, and we are still looking for answers on a huge number of key issues. The natural variability in streams and flows is related to natural cycles, but also to anthropogenic activities. For the Central European region, an increase in the frequency, duration, and severity of extreme hydro meteorological phenomena is expected in the future as a direct consequence of climate change. Climate change alters the interdependence structures of the hydrological variables. On the other hand, urbanization, channel regulation, dams, and many other interventions can influence the behavior of the basin during extreme hydrological events or affect the travel times of floods. The monitoring and evaluation of extreme hydrological phenomena using various models and methods is still necessary as a result of anthropogenic activities and climatic changes, which can negatively affect the application of frequency analysis. The statistical approach to hydrological analyses based on variables that occur only once per year represents the most frequently used approach in probabilistic hydrology. This approach has three limitations. The first is the length of the data series, which can frequently be less than 100 years. The second is the non-complete time-series data. Because they are incomplete, they may not provide comprehensive information about the conditions in the basin before the extreme flooding event occurs. The last one is the choice of suitable probability distribution functions, parameter estimation methods, and study period. An important factor in the correct estimation of extremes is the uncertainty of the applied statistical method. The estimation of the uncertainty in the designed discharges was investigated in [50] and [51]. For example, in our analysis, the four parametric JSB distribution fit the empirical data the best and increased the number of model parameters to nine. Stedinger et al. [52] preferred the generalized extreme value (GEV) distribution for estimating hydrological extremes. Millington et al. [53] examined the suitability of several types of probability distributions (GEV, LPIII, and Gumbel) for estimating the T-year discharges. We must not forget the fact that discharges with longer return periods (usually up 200 years) are extrapolated values burdened by systematic estimation errors as well as the choice of statistical methodology. A flood bivariate hydrological analysis of extremes plays an important role in flood risk analysis. Such analyses provide an overview of the flood event as a whole and can enable a more reliable assessment of flood risks and subsequent flood protection. The time variance

and asymmetry of copulas make them suitable tools for such analyses. The applications of the copulas in the frequency analysis of the hydrological extremes are very often focused on the interdependence analyses between hydrological variables such as discharge–volume, volume–duration, and discharge–duration. The copula function used as a mathematical tool in the frequency analysis of confluence flood waves was introduced in the work of Wang [54]. He applied four often-used Archimedean copulas to confluence floods and concluded that the Frank copula and the Gumbel–Hougaard copula are suitable statistical tools. The results of the bivariate analysis in Poyang Lake and the Yangtze River also showed that the coincidence probabilities were higher for flood events with shorter return periods [55]. However, the authors of [55] selected the Clayton copula as the better choice. In recent years, the use of the copula functions in the bivariate analysis of the flood risk at the river confluences has been increasing. Various authors (e.g., see [5,56–58]) dealt with the effect of synchronous extreme events on the mainstream and its tributaries. As we previously mentioned, the natural transformation of flood waves is also being increasingly affected and disturbs the artificial interventions in river basins and climate change. This can result in the synchronous occurrence of flood waves at the confluence of the rivers.

## 5. Conclusions

The paper presents an evaluation of the bivariate joint probability approach that may provide a practical method for performing the frequency analysis of floods at river confluences using a copula function. We focused on the statistical bivariate analysis of flood hazards at selected confluences of Slovak rivers.

The presented research showed the following:

- The copula-based joint probability approach for the confluence flood estimation performed well for the selected river basins;
- The copula-based joint probability approach provides a way to estimate the confluence flood without the discharge records needed for the mainstream below the confluence and without difficult computations such as flow routing;
- The copula functions for the multivariate analyses enable the use of various types of marginal distributions and thus release the limitation of the others in the case of multivariate approaches where the margins follow the same type of distributions. In our study, based on the selected criterions and the tests, the same type of probability distribution fit the analyzed data, except for Nitrianska Streda Station, situated below the Nitra–Bebrava confluence;
- The joint return periods calculated using copulas could be used to determine the severity of floods based on the desired relations between the mainstreams and their tributaries, looking for the exceedance of both variables.

Although this work was carried out in the basins of Slovakia, the methodology, despite its limitations, is also applicable for hydrological analyses in other localities. In our work, we used the longest possible series of input data, the availability of which in other (foreign) locations can be problematic. For this reason, we cooperate very closely with the countries of the Danube River region.

**Author Contributions:** Conceptualization, V.B.M.; Formal analysis, V.B.M.; Investigation, V.B.M., D.H. and P.P.; Writing—original draft, V.B.M.; Writing—review and editing, P.P., D.H. and P.M. All authors have read and agreed to the published version of the manuscript.

**Funding:** This research received no external funding.

**Data Availability Statement:** Data sharing is not applicable.

**Acknowledgments:** This work was supported by the project VEGA No. 2/0015/23 "Comprehensive analysis of the quantity and quality of water regime development in streams and their mutual dependence in selected Slovak basins"; project APVV-20-0374 "Regional detection, attribution and projection of impacts of climate variability and climate change on runoff regimes in Slovakia";

and project WATSIM "Water temperature simulation during summer low flow conditions in the Danube basin".

**Conflicts of Interest:** The authors declare no conflict of interest.

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
