# Peer review of "The Copula Application for Analysis of the Flood Threat at the River Confluences in the Danube River Basin in Slovakia"

_water, doi:10.3390/w15050984_

Round 1

Reviewer 1 Report

The copula application for analysis of the flood threat at the Slovak river confluences

This is an interesting study and I read the manuscript with interest.

Abstract: I would like to see the research gap for this work in the abstract.

Introduction: Authors have used many references to state this statement and I think the authors at least have to explain what these papers say rather than citing them as a bulk. “The Archimedean class of copula functions is very popular in hydrological applications for studying the mutual relations between the individual characteristics of a hydrological cycle [9-20].”

Wrong way of citing references. The study of the bivariate dependences and joint probabilities of various hydrological variables of the Morava River (Slovakia) using specific copulas was published in [21]. In [22] …

Many errors in references can be seen. You will have to correct them all and please see the way of writing and citing papers.

Figures, specially the plots are very unclear. You need to redraw them higher resolutions.

Figure 4 – You have scattered data when the probability is at 1. However, diverse points at lower probabilities. I think, you need to physically interpret these two cases and explain them to non-engineering background people too. Then only the take home message is taken from the research.

Reviewer 2 Report

The authors have presented a statistical technique for estimating extreme discharges and return periods at the confluences of rivers. The authors use flow data (maximum discharges) at 4 rivers and associated tributaries in Slovakia over a time frame of decades for their study. They use a few techniques, including a univariate statistical analysis and bivariate using Gumbel-Hougaard copula, which they present in the methods. Comparing these methods, they authors suggest that the copula method is useful for performing frequency analysis of floods. In reviewing this paper, the authors give a clear and solid presentation of their work, with necessary equations (including explanations and citations), supporting figures and tables, and conclusions. The writing is generally clear, although there are some instances where grammar could be improved to improve clarity of presentation. I appreciate the large data sets used and that the authors used such long time periods of data. There are some minor revisions that would improve the presentation of the paper. I recommend after some minor revisions that this would be suitable for publication in Water.

1.     For the equations, there are a lot parameters that readers may not be intrinsically familiar with. Be clear to define and describe all the variables so people don’t get lost in the equations.

2.     Since Water uses intext brackets for citations, it would be better to phrase sentences with that in mind. For example, line 204-205, the authors say “In [47], this probability distribution is recommended.” Rather you can say “Svanidze et al. recommend this probability distribution…[47].”

3.     The figures are fuzzy and difficult to read. This could be a function of creating a PDF out of the figure files. However, as it stands, they need to be improved and made crystal clear for publication.

4.     Figure 1, the map is too small and the fontsize of the legend is too hard to read.

a.     The figure would be improved by being about twice the size. Including a North marker and stating somewhere that this is a full map of Slovakia.

b.     The plots are fuzzy and could be larger. Also, an arrow pointing where the each figure is connected to on the map would be useful.

c.     The figure legend could have more details. Including that this is a map of Slovakia, with hydrographs of 4 locations and name those locations.

5.     Rest of the figures are similarly fuzzy. Consider making axis labels a bigger font.

6.     Figure 5. Not clear that histograms are the best for this. It is nearly impossible to discern what the values of R and Kendall are from this figures. A table may be a better presentation for only 8 numbers. If not, it might be better to make the figures smaller with the columns next to each other and have dashed lines from the y-axis values, or possibly 2D may be better than 3D.

7.     Figure 7. The legend or the figure should be clear that the left column is “or” and the right column is “and.” Perhaps even might be useful to explain why the shape of the curve is different for each of these figures.

8.     The second paragraph in Conclusions may be better suited for the Discussion. The Discussion digs more into how this work ties into other research, which is useful, but it may be good to discuss the results of this work first. Specifically, it would be useful to discuss how the statistics played out, and what it means in the context of your work.

9.     The authors discuss the limitations of the study, though I would also argue that the authors used relatively large data sets, which really helps bolster the work done.

10.  It may be worth pointing out in the Conclusions that even though this work was done in Slovakia, the methodology and utility of what you’ve presented can readily be picked up and moved to other locations. This isn’t a site-specific study, it is an applicable hydrology analysis study.

Reviewer 3 Report

The review of the manuscript title: “The copula application for analysis of the flood threat at the Slovak river confluences” written by Mitková and co-authors form Slovak Academy of Sciences.

The main problem analyzed in the presentred research is estimation of coincidences of maxium flows in the main streams and its tributaries in the selected rivers of Slovakia. The authors used the standard data from the long-term observations in the 12 guaga stations. The applied methods is the Gumbel–Hougaard copula functions. In general, the problem is important and interesting. However, the advancement of the applied methods is not impressive today, but the research may be worth to be published and extended in the future.

It is good to mention that structure of the text is consistent with classical structure of the scientific papers. The paper is well written and the provided explanations are clear enough. The authors carefully prepared introduction to the problem. The purpose of the research is precisely defined. They also presented quite interesting discussion of the obtained results. On the other hand, the approach to preparation of the conculsions should be a little bit different, what is explained below.

According to the reviewr evaluation the manuscript requires minor corrections before publication. These are described below.

Detailed remarks

1. Title of the paper

Maybe the title should be more precise. The authors analyze the confluences of selected rivers and their tributaries in the Danube river basin in the territory of Slovakia.

2.      Line 104:

One sentence of the comment on the expression “the least anthropogenic influence” could be added. As I understand the authors want to analyzed as natural conditions as possible. But it's also the specific of this research.

3.      Lines 105-107, Table 1:

In my opinion it could be really important what is the distance between each gauge station and the junction. The important information may be also the area of the watershed for each gauge station.

4.      Figure 1:

This scheme is not very clear. Please reconsider implementation of different map/maps.

5.      Lines 116-120:

If we take into account the fact that the surface part of the analyzed catchments should be independent systems, the dependency between the Qmaxup and Qmaxtr can be explained only by the coincidence of atmospheric conditions. In basic words, the coincidence exists only if the same rain falls in both catchments.

Is it correct intuition? Could you comment this?

6.      Figure 2:

The quality of this graphs have to be improved.

7.      Figure 3:

The titles of the graphs and descriptions of the axes are not clear.

8.      Figure 4:

I suggest the removal of the grid in the background of these graphs.

9.      Conclusions:

This section is written like a summary and extension of the Discussion. The summary may be a part of the Conclusions, but it should not be the major element. In this section the information about the importance of the presented achievements should be present. The authors should focus on the impact of the presented results on other areas of science and practice. The limitations of the study can be also explained. And finally the further steps could be designed.

Round 2

Reviewer 1 Report

Authors have taken a lead to fix their issues and therefore, I would like to accept this paper.
